# Comparative Analysis of AML Classification Systems: Evaluating the WHO, ICC, and ELN Frameworks and Their Distinctions

**DOI:** 10.3390/cancers16162915

**Published:** 2024-08-22

**Authors:** Huda Salman

**Affiliations:** Brown Center for Immunotherapy, Melvin and Bren Simon Comprehensive Cancer Center, School of Medicine, Indiana University, Indianapolis, IN 46202, USA; hsalman@iu.edu

**Keywords:** acute myeloid leukemia, AML, WHO, ICC, ELN, classification

## Abstract

**Simple Summary:**

The many classifications of AML add richness to what practitioners should focus on when treating their patients. However, the differences between these classifications, especially if they implicate a difference in the choice of therapy, may complicate clinical judgment. This review aims to point out these differences and examine how they affect therapeutic decisions.

**Abstract:**

Comprehensive analyses of the molecular heterogeneity of acute myelogenous leukemia, AML, particularly when malignant cells retain normal karyotype, has significantly evolved. In 2022, significant revisions were introduced in the World Health Organization (WHO) classification and the European LeukemiaNet (ELN) 2022 guidelines of acute myeloid leukemia (AML). These revisions coincided with the inception of the first version of the International Consensus Classification (ICC) for AML. These modifications aim to improve diagnosis and treatment outcomes via a comprehensive incorporation of sophisticated genetic and clinical parameters as well as facilitate accruals to innovative clinical trials. Key updates include modifications to the blast count criteria for AML diagnosis, with WHO 2022 eliminating the ≥20% blast requirement in the presence of AML-defining abnormalities and ICC 2022 setting a 10% cutoff for recurrent genetic abnormalities. Additionally, new categories, such as AML with mutated TP53 and MDS/AML, were introduced. ELN 2022 guidelines retained risk stratification approach and emphasized the critical role of measurable residual disease (MRD) that increased the use of next-generation sequencing (NGS) and flow cytometry testing. These revisions underscore the importance of precise classification for targeted treatment strategies and improved patient outcomes. How much difference versus concordance these classifications present and the impact of those on clinical practice is a continuing discussion.

## 1. Introduction

Acute myeloid leukemia (AML) encompasses a biologically diverse range of disorders characterized by the rapid and uncontrolled growth of clonal hematopoietic progenitor cells [1,2]. It is the most prevalent type of acute leukemia in adults, typically diagnosed at a median age of 68 years [3]. Historically, the overall survival (OS) rate for AML patients over five years has been around 30%, with significant differences observed across age groups [4]. Younger patients tend to have a better prognosis, with a 5-year OS rate near 50%, in contrast to less than 10% for those aged 60 and older [5]. However, these figures are primarily derived from older clinical trials and demographic data and may be revised considering the FDA approval of numerous novel drugs or combinations over the past decade (Figure 1).

AML is enriched with somatic mutations that can now be identified in 97.3% of cases [2]. As such, the biological foundation of AML has significant prognostic implications, which are further influenced by age and comorbid conditions. Advances in molecular understanding of AML’s evolution, hierarchical organization, kinetics, and resistance mechanisms have profoundly impacted diagnosis, risk stratification, and disease monitoring [6,7]. Therefore, accurate classification and risk stratification at diagnosis are crucial for determining the most effective therapeutic strategies.

As of now, there are three primary classification systems for AML that are widely recognized and used in clinical practice. WHO premiered and has revised its classifications over the years. These classifications have evolved as our understanding of AML and related myeloid neoplasms has improved, incorporating new molecular and genetic insights [8,9] (Table 1).

Additionally, an international coalition of pathologists and clinicians, many of whom contributed to previous editions of the WHO classifications, independently and concurrently organized the International Consensus Conference [10]. The International Consensus Classification (ICC) of myeloid neoplasms, introduced in 2022, represents a collaborative effort to create a globally accepted framework for classifying myeloid neoplasms, including AML. The ICC was developed to harmonize with other classification systems while incorporating the latest advances in molecular genetics and clinical research [7].

In addition to the WHO and the ICC input in AML classification, a third consensus, the ELN recommendations for the diagnosis and management of AML in adults, was published in 2010 [11] and similarly underwent repeated revisions [12,13]. These updates have primarily focused on refining risk stratification based on advances in genetic and molecular understanding, which directly impact prognosis and treatment decisions. ELN continues to emphasize the impact of MRD on treatment outcome [14,15].

The simultaneous adoption of multiple classification systems introduced new challenges regarding the intricacies of their clinical utility. Those challenges are mainly related to the interpretation and implementation of standardized treatment protocols. Hence, understanding their peculiarities and how they merge and differ is of exquisite importance.

## 2. Decoding AML Classification Systems

As shown in Table 1, the classification and diagnosis of AML underwent significant revisions with the introduction of the WHO 5th edition, the ICC first version, and the third revision of ELN Guidelines, known as WHO 2022, ICC 2022, and ELN 2022 classifications [6,7,13]. All three systems incorporated clinical, molecular/genetic, morphological, and immunophenotypic features and presented a robust proposal for the AML clinical framework.

**Updates on Blast Count**. WHO 2022 has eliminated the requirement of a ≥20% blast count as a diagnostic cut-off should a defining genetic abnormality such as PML/RARA in acute promyelocytic leukemia (APL) and core-binding factor AML (e.g., *RUNX1-RUNX1T1*, *CBFB-MYH11*) be present. They continued to require the 20% blast count for specific exceptions such as AML with (*BCR;ABL1*) fusion, *CEBPA* mutation, and other rarer genetic alterations. This change is based on evidence showing that patients with these certain genetic abnormalities and less than 20% blasts (formerly classified as myelodysplastic syndrome [MDS] under WHO 2017) have clinical outcomes comparable to those with higher blast percentages that qualified for the diagnosis of AML. For example, cases that were previously labeled as MDS or MDS/MPN with NPM1 mutations often rapidly progress to AML. Comparatively, ICC 2022 classification mandates a blast count cutoff of 10% for diagnosing AML with recurrent genetic abnormalities, except for AML with (*BCR;ABL1*) fusion, which still requires a 20% blast threshold. Both classifications continued to require a 20% blast requirement for AML with BCR:ABL1 fusion to prevent diagnostic intersection with the parent disease, chronic myeloid leukemia (CML).

**Employment of Vigorous Molecular Landscapes**. Both WHO 2022 and ICC 2022 classifications have expanded the list of genetic abnormalities and made specific changes. The genetic abnormalities to characterize AML in WHO 2022 and the recurrent genetic anomalies outlined in ICC 2022 are predominantly aligned with those described in WHO 2016, with only slight variations. For example, in the case of CEBPA mutations, the ICC 2022 classification focuses specifically on in-frame bZIP CEBPA mutations, while the WHO 2022 classification includes both biallelic (biCEBPA) and single mutations within the basic leucine zipper (bZIP) region of the gene (smbZIP-CEBPA). These updates are based on recent studies that highlighted the favorable prognosis associated with in-frame bZIP CEBPA mutations. Furthermore, both WHO 2022 and ICC 2022 have removed the provisional category of AML with mutated RUNX1, which was present in WHO 2017, due to lack of sufficient evidence supporting its status as a distinct entity. Figure 2 demonstrates concordance points and differences in AML-defining abnormalities.

A number of AML-defining balanced translocations that were used in WHO 2017 are notably absent from the updated WHO 2022 and ICC 2022 classifications, while ELN continue to incorporate those in AML risk stratification. This shift reflects the need to emphasize clinically relevant genetic abnormalities that more precisely guide diagnosis and treatment. Examples of translocations excluded are t(6;9), inv(3), or t(3;3), which were previously included under AML-MRC in the WHO 2017, might not be specifically categorized under AML-MRC in the new classifications. Instead, such genetic abnormalities are now considered within broader or different genetic contexts.

## 3. New Categories of AML in ICC 2022

**AML with Mutated *TP53*** has been added to the 2022 ICC classification. Somatic *TP53* mutations with a variant allele fraction exceeding 10% and a blast count of ≥20% defined this AML. Cases with 10–19% blasts are classified as MDS/AML with mutated *TP53*. This category now also includes TP53 mutated pure erythroid leukemia (PEL).

**MDS/AML** has been introduced, replacing the former WHO 2017 classification of MDS with excess blasts. This new classification includes cases with 10–19% blasts in the peripheral blood or bone marrow. The designation of MDS/AML enables patients within this blast range to be included in clinical trials for either MDS or AML, depending on their specific clinical circumstances. The diagnostic criteria for MDS/AML remain consistent with those for AML requiring ≥20% blasts.

In contrast, the WHO 2022 classification continues to categorize these patients as MDS-IB2 (increased blasts, grade 2), a term updated from the previous “excess blasts” to minimize the risk of overtreatment. However, WHO 2022 acknowledges that MDS-IB2 can be treated as AML if the clinical situation warrants such an approach.

## 4. New Terminology in WHO 2022

**AML defined by differentiation and AML not otherwise specified**. For cases with blast counts ≥ 20%, they are now classified as AML defined by differentiation according to WHO, and as AML, not otherwise specified (NOS), by ICC. Both systems extrapolated these terminologies from the WHO 2017 classification.

**Acute erythroid leukemia** (AEL) replaces pure erythroid leukemia in the WHO 2022. To qualify for AEL, erythroid predominance, defined as ≥80% of bone marrow elements, with ≥30% proerythroblasts (or pro-normoblasts) must be present.

**AML, myelodysplasia-related (AML-MR)**. AML with myelodysplasia-related changes (AML-MRC) in WHO 2017 is AML, is myelodysplasia-related (AML-MR) in WHO 2022. The ICC 2022 further divides this entity into two categories, AML with myelodysplasia-related gene mutations and AML with myelodysplasia-related cytogenetic abnormalities. The former include *ASXL1*, *RUNX1*, *SETBP1*, *SRSF2*, *STAG2*, *U2AF1*, *ZRSR2*, and *BCOR*. Those with cytogenetics abnormalities include complex karyotype, -7 or del(7q), -5 or del(5q), i(17q) or t(17p), -13 or del(13q), del(11q), del(12p) or t(12p), and idic(X)(q13). Both classifications mandate a threshold of ≥20% myeloblasts for this AML. AML-MR now combines specific genetic abnormalities and/or a prior history of myelodysplastic syndromes (MDS) or myelodysplastic/myeloproliferative neoplasms (MDS/MPN). In ICC 2022, a history of MDS or MDS/MPN serves as a descriptive element; myeloid neoplasm post-cytotoxic therapy, myeloid neoplasms with associated germline predisposition, and myeloid proliferation associated with Down syndrome.

**Updated Classification of Secondary Myeloid Neoplasms**. Three entities of secondary myeloid leukemias are now included in the WHO 2022 classification. These are myeloid neoplasm post-cytotoxic therapy, myeloid neoplasms with associated germline predisposition, and myeloid proliferation associated with Down syndrome.

## 5. Update on Secondary Myeloid Neoplasms

The WHO 2022 classification emphasizes the distinctiveness of secondary myeloid neoplasms, which include secondary AML, secondary MDS, and secondary MDS/MPN. WHO 2022 categorizes secondary myeloid neoplasms based on their association with prior exposure to chemotherapy or radiation (therapy-related) or as arising from a pre-existing myeloid disorder, such as MDS or MPN. The WHO classification continues to highlight the importance of genetic mutations, particularly those that are commonly associated with therapy-related neoplasms, such as TP53, RUNX1, complex karyotypes, and specific translocations and deletions (e.g., t(6;9), t(9;11)), and deletions (e.g., del(5q), del(7q)) are associated with secondary myeloid neoplasms and are integrated into the classification scheme. Lastly, the classification places a strong emphasis on the clinical history of the patient, particularly any prior hematologic disorder or exposure to cytotoxic therapies, to appropriately categorize the myeloid neoplasm as secondary.

The ICC 2022 provides a refined classification that is closely aligned with advances in molecular diagnostics and the clinical relevance of these classifications. For secondary myeloid neoplasms, ICC 2022 highlights specific mutations indicative of therapy-related myeloid neoplasms or those arising from pre-existing conditions such as TP53, ASXL1, EZH2, and DNMT3A. The prognostic impact of co-mutations such as FLT3 co-occurrence with TP53 is used for prognostic evaluation. The ICC emphasizes the importance of these classifications in guiding treatment decisions, recognizing that secondary myeloid neoplasms often have distinct biological and clinical characteristics compared to their de novo counterparts. The classification makes a clear distinction between therapy-related and other secondary myeloid neoplasms, particularly in the context of the genetic and molecular alterations that define them.

## 6. Revisions in the ELN 2022

**ELN updated its risk stratification strategy for AML**. The ELN 2017 updated its risk stratification to align with the ICC classification, resulting in the new ELN 2022 guidelines. Significant changes have been implemented within the stratification of favorable, intermediate, and adverse risks AML. Table 2 details the ELN updated risk stratification [13].

***FLT3-ITD* Allelic Ratio**: the *FLT3-ITD* allelic ratio is no longer used in risk classification. Consequently, AML with *FLT3-ITD* (in the absence of other adverse-risk genetic lesions) is now classified as intermediate risk, irrespective of the allelic ratio or the presence of an *NPM1* mutation [13]. This change addresses the challenges in standardizing the *FLT3-ITD* allelic ratio measurement across different assays, the modifying effects of *FLT3* inhibitor-based therapy that improved survival regardless of allelic ratio, and the increasing importance of minimal/measurable residual disease that now offers a more precise and actionable metric for tailoring therapy in treatment decisions. Additionally, the elimination of the allelic ratio allowed treatment regimens to be more consistently applied, reducing variability and potential disparities in patient care.

**ELN Expands Adverse Prognosis Genes**: genes associated with adverse prognosis have been expanded to include *TP53*, a tumor suppressor gene that plays a crucial role in regulating the cell cycle and preventing genomic instability [16], *ASXL1* [17], *BCOR* [18], and *EZH2* [19], play key roles in chromatin modification, *RUNX1* [20] transcription, *SF3B1* [21], *SRSF2* [22], *U2AF1* [23], and *ZRSR2* [24], RNA splicers and regulators and *STAG2* [25], which plays a key role in chromosome segregation and cohesion.

**ELN 2022 Updates Molecular Features in Secondary Myeloid Neoplasms**. The ELN 2022 guidelines stratify risk in secondary myeloid neoplasms by including key mutations such as TP53, ASXL1, and RUNX1. These mutations are considered high-risk factors, and their presence can influence the decision to pursue more aggressive treatment options, such as allogeneic stem cell transplantation. The presence of complex cytogenetics, often associated with TP53 mutations, is integrated into the risk models as well.

**ELN Continues to Highlight the Impact of MRD**. MRD detection and subsequent monitoring are crucial for assessing the effectiveness of therapy, predicting relapse, and guiding subsequent treatment decisions, including the potential need for stem cell transplants [15,26]. The ELN 2022 guidelines underscore the critical importance of monitoring MRD using molecular techniques such as flow cytometry and NGS, particularly in cases with specific actionable mutations. While modern flow cytometry techniques offer high sensitivity and specificity [27], NGS is encouraged to be part of MRD detection as well [28].

Molecular MRD detection is particularly valuable for its prognostic implications. Patients with detectable MRD post-treatment have a higher risk of relapse and poorer overall survival compared to those who achieve MRD negativity. MRD detection helps in tailoring treatment strategies. For instance, patients with MRD positivity might benefit from additional chemotherapy, targeted therapy, or enrollment in clinical trials exploring novel treatments. MRD status can influence the decision to proceed with allogeneic stem cell transplantation [29]. Patients with persistent MRD after initial treatment are often considered for transplantation due to the higher risk of relapse. MRD persistence before transplant is associated with poorer transplant outcomes compared to MRD-negative transplants. Additionally, persistent or recurrent MRD after transplantation can prompt preemptive interventions to prevent overt relapse.

MRD status is often used as a criterion for enrollment in clinical trials. Patients with MRD positivity may be candidates for trials investigating new drugs or treatment approaches aimed at eradicating residual disease.

## 7. Personalized Treatment Strategies Enabled by the Revised Classifications

Complementing the parent WHO classification, ICC and ELN classifications provide a detailed genetic and molecular framework that significantly enhances the ability to personalize treatment strategies for AML. These classifications allow clinicians to tailor therapy to specific genetic abnormalities, optimizing outcomes and minimizing unnecessary toxicity. Patients are stratified into different risk categories based on their genetic profiles, which guides the intensity of their treatment. For example, patients with favorable risk features (e.g., *NPM1* mutation without *FLT3-ITD*) [30] may receive less intensive consolidation therapy, whereas those with adverse risk features (e.g., *TP53* mutation, complex karyotype) [31] may be directed towards more aggressive treatments like allo-SCT.

**TP53 Mutated AML**. Both ICC and ELN highlight the poor prognosis associated with *TP53* mutations, often necessitating more aggressive treatment strategies such as early consideration for allogeneic stem cell transplantation (allo-SCT) due to the high risk of relapse and resistance to conventional chemotherapy.

**Favorable prognosis mutations like *NPM1***, when not accompanied by *FLT3-ITD* or other high-risk mutations, often allow for less aggressive consolidation strategies and potentially avoid allo-SCT in the first remission.

***FLT3-ITD***, particularly with a high allelic ratio, is associated with poor prognosis and the need for targeted therapies such as *FLT3* inhibitors (e.g., midostaurin, gilteritinib) along with standard chemotherapy [29].

***IDH1/2* inhibitors** (e.g., ivosidenib, enasidenib) are used for patients with *IDH1* or *IDH2* mutations [32].

**Eligibility for allo-SCT**. The choice of post-remission therapy, including the need for allo-SCT, is heavily influenced by MRD status and genetic risk. MRD-negative patients with favorable genetics may continue with conventional consolidation, whereas MRD-positive patients or those with high-risk genetics may proceed to allo-SCT to reduce the risk of relapse. MRD monitoring through advanced techniques such as flow cytometry, PCR, and NGS is emphasized in both ICC and ELN classifications. Achieving MRD negativity is a crucial goal, as it is associated with significantly better long-term outcomes. MRD-positive patients, even if in morphological remission, may require additional or intensified treatment to prevent relapse.

**Clinical Trial Eligibility** and **Stratified Enrollment**. Detailed genetic and molecular profiling enables more precise stratification of patients for clinical trials. This ensures that new therapies are tested on the most appropriate patient populations, potentially leading to more effective and personalized treatment options being developed and approved.

**Facilitation of Treatment Adjustments**. Continuous monitoring of MRD allows for dynamic adjustments in treatment [33]. For instance, if a patient initially achieves MRD negativity but later shows MRD positivity, additional treatment can be administered to prevent full relapse. This approach maximizes the chances of maintaining long-term remission.

## 8. Impact of Classification Variations on AML Diagnosis and Treatment

The updated classifications have significantly advanced the understanding and management of AML by incorporating more refined approaches based on molecular and genetic features. The integration of these molecular and genetic characteristics has enabled clinicians to achieve greater diagnostic precision, enhance risk stratification, and customize treatment strategies to align with individual patient profiles. This evolution in classification has facilitated the adoption of more personalized and targeted therapeutic interventions, leading to improved prognostic accuracy and better patient outcomes. However, variations among these classification systems may necessitate careful interpretation and application in clinical practice.

Except for the consistent requirement of a 20% blast count in cases with (BCR; ABL) translocations in both the WHO and ICC classifications, the criteria for AML-defining abnormalities concerning blast count, for example, diverge between the two systems. The ICC sets a 10% blast count threshold, while the WHO has eliminated a specific blast count requirement, allowing for a diagnosis without a minimal blast percentage. Additionally, trisomy 8, del(20q), and RUNX1 mutations are recognized as AML-defining in the ICC, but not in the WHO classification. Conversely, the WHO identifies 11q deletions and monosomy 13 as AML-defining, whereas the ICC does not. Balanced chromosomal translocations, which serve as risk stratifiers in the ELN, have been excluded from both the WHO and ICC classifications. The challenge for practicing physicians lies in reconciling these differences, as they may impact management decisions by altering diagnostic criteria. This issue warrants further discussion within the medical community. As of now, very few reported on the impact of these variations among the classification systems. To evaluate this impact, Huber et al. [34] analyzed 717 MDS and 734 non-therapy-related AML patients, initially diagnosed according to the WHO 2017, using whole genome and transcriptome sequencing. The analysis reveals a reduction in purely morphologically defined AML entities from 13% to 5% under both new classifications. Myelodysplasia-related (MR) AML increased from 22% to 28% (WHO 2022) and 26% (ICC). Genetically defined AML remained the largest category, with the previously recognized AML-RUNX1 predominantly reclassified as AML-MR (WHO 2022: 77%; ICC: 96%). Differences in the inclusion criteria for AML-*CEBPA* and AML-MR, such as the exclusion of *TP53*-mutated cases in the ICC, were linked to variations in overall survival outcomes [34]. This work demonstrates that both classifications emphasize genetics-based definitions yet share similar foundational concepts and a high degree of concordance. However, remaining discrepancies, particularly in cases like *TP53*-mutated AML, necessitate further studies to resolve outstanding questions in disease categorization. Thus, and based on limited available comparative studies, the benefit of a rather unified classification system is up for continuing review at this time.

## 9. MDS-Related Gene Mutations and MDS-Related Cytogenetic Abnormalities as Updated in WHO and ICC 2022

WHO 2022 tends to maintain traditional cytogenetic classifications while integrating new genetic mutations into the existing framework. It allows for the broader inclusion of genetic abnormalities within its subtypes. ICC 2022 focuses more on integrating specific genetic mutations with cytogenetic abnormalities, often reclassifying cases with high-risk genetic profiles (like TP53 mutations) into higher-risk categories. It emphasizes a more clinically relevant approach, sometimes leading to differences in how certain abnormalities are classified compared to WHO.

Both classifications share a significant overlap but differ in how strictly they interpret and combine genetic and cytogenetic data for defining MDS subtypes and risk categories.

***SF3B1* Mutation**. WHO 2022 defines a specific subtype, “MDS with mutated *SF3B1*”, characterized by *SF3B1* mutations. This subtype is typically associated with ring sideroblasts and a relatively favorable prognosis. ICC 2022 also recognizes “MDS with *SF3B1* mutation” as a distinct entity. Like WHO, ICC focuses on the presence of ring sideroblasts and classifies these cases based on the mutation’s presence and percentage of blasts. However, the ICC may have different criteria for the number of ring sideroblasts required for this classification.

***TP53* Mutation**. In the WHO 2022 classification, *TP53* mutations are recognized as an important prognostic factor. Cases with *TP53* mutations are often considered higher risk, particularly when associated with complex karyotypes. The WHO classification may include *TP53*-mutated cases within various subtypes of MDS, depending on other concurrent abnormalities. In the ICC 2022 classification, *TP53*-mutated MDS is treated with special caution. The ICC has a more nuanced approach, sometimes excluding *TP53*-mutated cases from lower-risk categories, such as MDS with low blast count (MDS-LB), particularly if these mutations are associated with complex karyotypes. *TP53* mutations are often placed in high-risk categories regardless of other features due to their poor prognosis.

***ASXL1* Mutation**. WHO 2022 recognized the *ASXL1* mutation as a mutation that is frequently associated with poor prognosis. It is used as part of the overall assessment but does not define a specific subtype. Similar to WHO, *ASXL1* mutations are considered high-risk and are used in the overall risk stratification. ICC uses these mutations more explicitly in defining risk, especially in combination with other mutations.

***RUNX1* Mutation**. *RUNX1* mutations were previously associated with a distinct subtype in older WHO classifications but are now generally categorized within broader MDS groups. In ICC, *RUNX1* mutations are treated similarly to WHO 2022, focusing more on the mutation’s impact on prognosis rather than defining a separate subtype.

***CEBPA* Mutation**. WHO 2022 considers *CEBPA* mutations within the context of AML with myelodysplasia-related changes (AML-MRC) if present. In MDS, they may be mentioned but are not a major defining feature. ICC includes *CEBPA* mutations in the classification but may exclude cases with concurrent *TP53* mutations from certain categories, similar to its approach with *TP53* mutations.

**del(5q)**. WHO 2022 defines a specific subtype, “MDS with isolated del(5q)”, which is associated with a relatively favorable prognosis. This is a well-established category in the WHO classification. ICC 2022 also recognizes “MDS with isolated del(5q)” as a distinct entity. The ICC’s criteria are similar to the WHO’s but may place additional emphasis on concurrent mutations (e.g., *TP53*) that can affect prognosis.

**Complex Karyotype**. Complex karyotype (three or more chromosomal abnormalities) is recognized as a marker of high-risk MDS in both the WHO and ICC classifications.

**-7/7q-**. The loss of chromosome 7 or deletion of 7q is considered a high-risk feature in MDS by both classifications. It is often associated with poor prognosis and is used in both the classification and risk stratification.

**+8 (Trisomy 8)** is recognized as a recurrent cytogenetic abnormality in MDS. It is considered in risk stratification but does not define a specific MDS subtype in either ICC or the WHO classifications.

**Inv(3)/t(3;3)** is considered as part of MDS with myelodysplasia-related changes (MDS-MRC), particularly when associated with other high-risk features in both classification systems.

## 10. Conclusions and Future Direction

The ongoing refinement of AML classification systems underscores the dynamic nature of cancer genomics and the critical importance of integrating emerging research into clinical practice. The 2022 revisions have advanced our understanding of AML by incorporating genetics-based definitions, leading to a more precise and personalized approach to diagnosis, risk stratification, and treatment. While these systems share a high degree of agreement in their foundational concepts, differences in the criteria, particularly concerning blast count thresholds and the inclusion of specific genetic abnormalities, may influence diagnostic and therapeutic decisions.

The review highlights two key factors that could significantly impact treatment outcomes: the blast count thresholds that define AML diagnosis in different molecular contexts and the rigor of minimal residual disease (MRD) detection. The WHO’s decision to eliminate a blast count cutoff in the presence of AML-defining abnormalities is a critical shift, emphasizing the prognostic significance of molecular profiles over traditional morphological criteria. This approach aligns with evidence suggesting that molecular characteristics, rather than blast count alone, should guide treatment strategies.

Future efforts should focus on standardizing MRD detection protocols across laboratories to ensure consistency and reliability, enhancing the integration of MRD monitoring into routine clinical practice. Additionally, as more genetic abnormalities become targetable, the practical impact of how these molecular markers are utilized in AML classification will grow, potentially leading to further revisions and updates in classification systems.

Ultimately, the evolving landscape of AML classification and management will require ongoing research, clinical trials, and collaborative efforts to ensure that these advances translate into improved patient outcomes. The dialogue within the medical community regarding the implications of these classification differences will be essential as we continue to refine and optimize treatment strategies for AML.

## Figures and Tables

**Figure 1 cancers-16-02915-f001:**
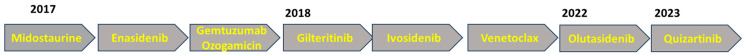
Timeline of FDA approvals for AML.

**Figure 2 cancers-16-02915-f002:**
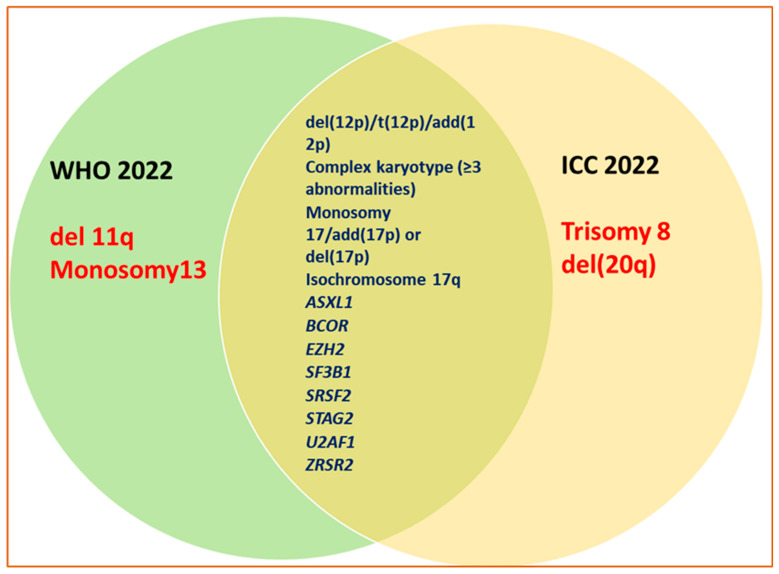
AML-defining abnormalities in both WHO and ICC 2022.

**Table 1 cancers-16-02915-t001:** History of AML classifications and key incorporations.

Classification System	Versions	Key Features
WHO	2001	- Emphasis on morphology, cytogenetics - Specific genetic translocations such as t(8;21)(q22;q22.1), inv(16)(p13.1q22), and t(15;17)(q24;q21) were recognized as distinct AML subtypes with specific clinical features and prognostic implications
	2008	- Inclusion of AML with NPM1 mutation - Inclusion of AML with CEBPA mutation - Refined the criteria for AML with multilineage dysplasia (AML-MDS) - Details on therapy-related AML, specifying different subtypes - Clarification and expansion of AML not otherwise specified (AML NOS)
	2016	- Expanded genetic and molecular abnormalities: mutated RUNX1, AML with BCR-ABL1 - Categorized therapy-related AML (t-AML) under the broader category of “therapy-related myeloid neoplasms” - Refined myelodysplastic/myeloproliferative neoplasms (MDS/MPN) and mixed-phenotype acute leukemia (MPAL)
	2022	- Refinement of genetic entities. Genes like NPM1, CEBPA, RUNX1, and others are based on updated research on their clinical significance - New recognized entities - Revised diagnostic criteria, updated blast count, and AML-defining abnormalities. - Terminology updates - Focus on actionable mutations
ICC	2022 (single version)	- Aims for broader clinical relevance and universal applicability - Distinct AML subtypes based on molecular integrations. NPM1, CEBPA, TP53, FLT3, and others - Alignment with other classification systems - Flexible framework for future advances
ELN	2010	- Primarily a risk stratification system rather than a detailed classification - Divides AML into favorable, intermediate 1 and 2, and adverse risk groups based on cytogenetic including t(8;21), inv(16), and t(15;17) for favorable risk, and complex karyotype or monosomal karyotype for adverse risk
	2017	- Continued the risk stratification framework, with favorable, intermediate, and adverse - Expanded genetic markers, RUNX1, ASXL1, and TP53 - FLT3-ITD refinements. A high allelic ratio of FLT3-ITD was classified as adverse, while a low ratio was considered intermediate when combined with other genetic factors
	2022	- The classification recognized the increasing importance of next-generation sequencing (NGS) in identifying genetic mutations that impact prognosis - New genetic markers such as DTA (DNMT3A, TET2, ASXL1) mutations were considered, particularly in relation to clonal hematopoiesis, and their role in risk stratification was further clarified - Personalized treatment emphasis.

**Table 2 cancers-16-02915-t002:** ELN 2022 [13].

Risk Category	Genetic Abnormality
Favorable	t(8;21) (q22;q22.1)/*RUNX1*::*RUNX1T1* inv(16)(p13.1q22) or t(16;16)(p13.1;q22)/*CBFB*::*MYH11* Mutated *NPM1* wihout *FLT3*-ITD bZIP in-frame mutated *CEBPA*
Intermediate	Mutated *NPM1* with *FLT3*-ITD Wild-type *NPM1* with *FLT3*-ITD (without adverse-risk genetic lesions) t(9;11) (p21.3;q23.3)/*MLLT3*::*KMT2A* Cytogenetic and/or molecular abnormalities not classified as favorable or adverse
Adverse	t(6;9)(p23.3;q34.1)/*DEK*::*NUP214* t(v;11q23.3)/*KMT2A*-rearranged t(9;22)(q34.1;q11.2)/*BCR*::*ABL1* t(8;16)(p11.2;p13.3)/*KAT6A*::*CREBBP* inv(3)(q21.3q26.2) or t(3;3)(q21.3;q26.2)/*GATA2*, *MECOM(EVI1)* t(3q26.2;v)/*MECOM*(*EVI1*)-rearranged −5 or del(5q); −7; −17/abn(17p) Complex karyotype, monosomal karyotype Mutated *ASXL1*, *BCOR*, *EZH2*, *RUNX1*, *SF3B1*, *SRSF2*, *STAG2*, *U2AF1*, and/or *ZRSR2* Mutated *TP53*

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
