# Peer review of "Comparative Analysis of AML Classification Systems: Evaluating the WHO, ICC, and ELN Frameworks and Their Distinctions"

_cancers, 2024, doi:10.3390/cancers16162915_

Round 1

Reviewer 1 Report

Comments and Suggestions for Authors

It is a very well written and comprehensive manuscript commenting on the differences and similarities between the two recently introduce classification system for AML.

Below there a few proposals for further improving the quality of the manuscript

1) it will be very helpful to the reader to add a Figure or a Table briefly presenting the ICC and the WHO-2022 classification systems

2) Authors should descibe or name its one of the AML-defining genetic abnormalities for ICC and WHO classification respectively

3) Authors should describe the differences between the two systems  regarding the MDS-related gene mutations and MDS-related cytogenetic abnormalities 

Author Response

Thank you for your valuable input. 

additional tables were added per recommendation.

MDS molecular features are also added. 

Reviewer 2 Report

Comments and Suggestions for Authors

Salman reviewed updated acute myeloid leukemia classifications, underlining differences between WHO and ICC and specifying the implications in terms of clinical management of patients. The review article is well-written, balanced and comprehensive, the quoted references overall adequate.

The authors have reported a summarizing table, the only thing I might suggest is a graphic figure depicting the succession of AML classifications over the years.

I have some minor points to raise that are listed below.

Minor issues

- Gene names should be reported in italics

- The nomenclature provides “::” per fusion genes: “BCR::ABL1 fusion”, please correct.

Author Response

Thank you for your valuable input. Table 1 is added to describe the history of AML classification over time. Other edits are also completed.

Reviewer 3 Report

Comments and Suggestions for Authors

There is redundancy in the text like "AML is a heterogenous disease" as is on lines 20 and 32 and many other ones which could be avoided. 

Manuscript is pretty descriptive and should have more detailed and specific discussion if at all and how these discrepancies influence treatment options. What are the practical consequences for the clinical hematologists? 

How should hematopathologists report this different classifications is a big issue and it could be written more about WHO 2022, not only ELN and ICC and a proposition for possible unification of the classifications..  

Comments on the Quality of English Language

se above

Author Response

Very insightful input to clarify more on the practical impact of these classifications on treatment options, and even on how the path report should be written. I expanded on that with also a personal opinion since this impact is being currently reviewed and studied by the medical community. some of this analysis is reported and I incorporated that. 

I am not sure how best is path reports reported other than to specify whay system was followed. Asking pathologist to compare should they have followed another system with what they follow to report is a bit premature since the classificatios are mostly concordent. However some aspects I agree withthe reviewr could be critical in the path report and my main issue is with the blast count relevent to different defining alterations. I tried to analyse this as well. open to suggestions. 

Revised the manuscript to minimize redundancy, again thank you for pointing this out. 

Round 2

Reviewer 3 Report

Comments and Suggestions for Authors

No further comments